# Secondary motor integration as a final arbiter in sensorimotor decision-making

**Tarryn Balsdon**[1]*, **Stijn Verdonck**[2], **Tim Loossens**[2], **Marios G. Philiastides**[1]*

**1** School of Psychology and Neuroscience, University of Glasgow, Glasgow, United Kingdom, **2** Faculty of Psychology and Educational Sciences, KU Leuven, Leuven, Belgium

* tarryn.balsdon@glasgow.ac.uk (TB); marios.philiastides@glasgow.ac.uk (MGP)

## Abstract

Sensorimotor decision-making is believed to involve a process of accumulating sensory evidence over time. While current theories posit a single accumulation process prior to planning an overt motor response, here, we propose an active role of motor processes in decision formation via a secondary leaky motor accumulation stage. The motor leak adapts the "memory" with which this secondary accumulator reintegrates the primary accumulated sensory evidence, thus adjusting the temporal smoothing in the motor evidence and, correspondingly, the lag between the primary and motor accumulators. We compare this framework against different single accumulator variants using formal model comparison, fitting choice, and response times in a task where human observers made categorical decisions about a noisy sequence of images, under different speed–accuracy trade-off instructions. We show that, rather than boundary adjustments (controlling the amount of evidence accumulated for decision commitment), adjustment of the leak in the secondary motor accumulator provides the better description of behavior across conditions. Importantly, we derive neural correlates of these 2 integration processes from electroencephalography data recorded during the same task and show that these neural correlates adhere to the neural response profiles predicted by the model. This framework thus provides a neurobiologically plausible description of sensorimotor decision-making that captures emerging evidence of the active role of motor processes in choice behavior.

## Introduction

The study of sensorimotor decision-making is currently enjoying an ever-increasing amount of attention across different species and levels of neuronal organization [1–5]. Central to these endeavors is a computational framework in which noisy sensory information is accumulated over time to an internal decision boundary [6–8]. Traditionally, this framework assumed that the decision boundary is independent of the amount of sensory evidence driving the decision and that once the boundary is reached, a choice is categorically communicated to the motor system to execute an overt response.

More recent evidence, however, appears to challenge both of these accounts. For instance, several animal and human electrophysiology studies have reported temporal accumulation profiles terminating at different boundaries, scaling proportionally to the amount of available

**Data Availability Statement:** Data availability All data (behavioral and EEG) are available to download from the Open Science Framework

(OSF) here: https://osf.io/xky4v/ (DOI 10.17605/OSF.IO/XKY4V). Code availability Analysis code is available to download from the Open Science Framework (OSF) here: https://osf.io/dwugq/ (DOI 10.17605/OSF.IO/DWUGQ).

**Funding:** This work was supported by the European Research Council (865003; to MGP), the Economic and Social Research Council (ES/L012995/1; to MGP), the Research Fund of KU Leuven (C14/19/054; to SV) and FWO (G074219N; to SV). The funders had no role in study design, data collection and analysis, decision to publish, or preparation of the manuscript.

**Competing interests:** The authors have declared that no competing interests exist.

**Abbreviations:** BIC, Bayesian information criterion; DDM, drift diffusion model; EEG, electroencephalography; LIT, Leaky Integrating Threshold; PCA, principal component analysis; ROC, receiver operator characteristic; SAT, speed/accuracy trade-off.

evidence (i.e., with task difficulty [9–14]). Similarly, a wide body of evidence appears to contradict the strict temporal dichotomy between decisional and motor processes and suggests that the (pre)motor system might play a more direct and causal role in decision-making [15–21].

Multiple neurophysiological studies have described activity related to motor preparation beginning before sensory evidence integration completes, effectively lagging the primary process of evidence accumulation [22–24]. This would allow the amount of sensory evidence to have a direct impact on motor planning, which is not accounted for in traditional evidence accumulation models. While some recent models have captured the build-up of motor preparation signals using inhibitory ("Gated Accumulator Model" [25,26]) or urgency ("Urgency Gating Model" [27–29]) gating, they do not preserve a separate sensory evidence accumulation signal and thus do not explicitly capture the entanglement of sensory evidence accumulation with motor preparation.

Here, we propose an alternative computational framework [30] that models this entanglement by introducing a secondary, motor-related, leaky integration process that receives the integrated evidence of the primary decision process as a continuous input and triggers the actual response when it reaches its own threshold. In other words, the primary evidence accumulator relinquishes control of the eventual choice (and hence the strict requirement of an evidence-independent decision boundary) by passing the integrated evidence along to the motor system.

We arbitrate between this alternative theoretical account and conventional single-integration models—including a variant with collapsing decision boundaries [31–34]—by leveraging computational modeling and time-resolved electroencephalography (EEG) data. We put a special emphasis on a task with a speed/accuracy trade-off (SAT) manipulation since the SAT offers an intuitive opportunity to exploit the proposed interplay between the primary and motor accumulators.

In doing so, we validate the newly proposed framework and simultaneously demonstrate that the SAT is likely primarily controlled by changes in the proposed leaky motor integration process rather than the long-standing view of boundary adjustments [8,35]. Our findings help reconcile the emerging inconsistencies in explaining SAT, most notably from animal electrophysiology experiments [36,37], and provide a foundation upon which future studies can continue to interrogate the neural systems underlying rapid sensorimotor decision-making.

## Results

In this work, we employed a speeded face-versus-car categorization task (Fig 1A; [11,38,39]) while multichannel EEG data were being collected. The task was designed specifically to induce a time-dependent accumulation of sensory evidence based on concrete perceptual categories through rapidly updating dynamic stimuli (see Materials and methods for more details). We used 2 levels of sensory evidence (low and high), by manipulating the phase coherence of the stimuli. We also introduced a SAT manipulation by controlling the amount of time participants had at their disposal to make a response across different blocks of trials (Speed blocks: 1 s versus Accuracy blocks: 1.6 s). Participants ($N = 43$) indicated their choice via a button press using the index finger of either their right or left hand. The mapping between face/car choices and left/right button presses was counterbalanced across participants. Mean response times were longer in the Accuracy condition (population average of 873 ms) than in the Speed condition (654 ms), while behavioral performance was markedly worse in the Speed condition (population average accuracy of 0.78) compared to the Accuracy condition (0.84) (Fig 1B; green and blue lines, respectively). Finally, for both the Speed and Accuracy conditions, mean response times were longer (by 101 ms, on average) and performance was worse (by 0.15, on

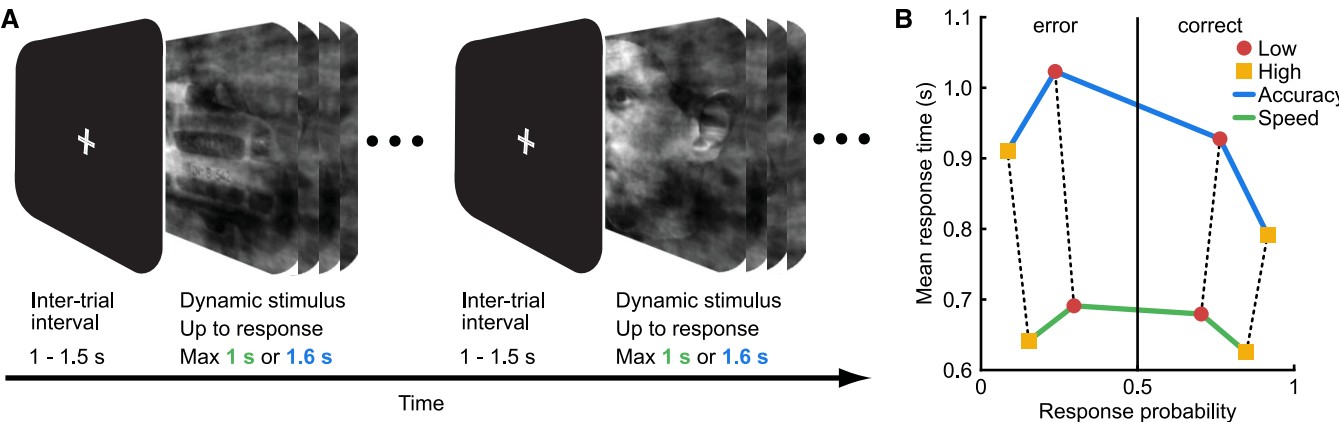

**Fig 1. Schematic representation of the experimental task and behavioral performance.** (a) Participants had to categorize dynamically updating sequences of noisy images as either a face or a car. In Speed trials, participants had up to 1 s to decide, while in Accuracy trials, they had up to 1.6 s. They indicated their choice with either a left or right hand button press, based on a predefined (participant-specific) mapping between the perceptual categories and the 2 hands. See text for more details. (b) Participant averaged latency–probability plot of face versus car categorization responses, for Accuracy (blue line) and Speed trials (green line), respectively. Mean response times are plotted on the left for the error responses (lower probability) and on the right for correct responses (higher probability). Mean response times related to the high evidence condition are denoted by the yellow symbols and to the low evidence condition by red symbols.

average) for the low compared to high evidence trials (Fig 1B; red circles and yellow squares, respectively).

## The leaky integrating threshold

The current generation of decision-making models make a number of fundamental assumptions. Invariably, the presentation of a stimulus is assumed to trigger and drive a process of evidence accumulation in which stimulus evidence is integrated over time. This accumulating evidence is interpreted as a decision variable that typically evolves toward a choice option appropriate for the stimulus that is being presented. A choice is made when the decision variable reaches some criterion or reference state. Finally, this choice is conveyed to the motor system, which, in turn, delivers the physical response.

In Verdonck and colleagues [30], we break with this tradition and propose to broadcast the accumulated evidence into a secondary, leaky motor accumulator and assume that the stopping criterion applies only for this secondary stage of the decision-making process. In other words, there is no longer a binary choice stemming from the initial evidence accumulation (which then triggers a motor response) but rather from a secondary motor accumulation process that simultaneously accumulates to its own threshold by taking the evidence emerging from the first process as input. This "Leaky Integrating Threshold" (LIT) effectively entangles evidence and motor accumulation, resulting in a host of specific predictions that can be falsified using both behavioral and neural data. The LIT can be applied to any model of decision-making, but in this paper, we use it to extend the common constant drift diffusion model (DDM).

Mathematically, the dynamics of the deterministic LIT are defined as

$$dy(t) = (\beta x(t) - \lambda y(t))dt \tag{1}$$

with $x(t)$ the evidence accumulated over time and $y(t)$ the resulting motor accumulation. For the analyses in this paper, we will use a simple constant drift diffusion process for the dynamics of $x(t)$ with stimulus-dependent drift speeds $v_i$ ($i$ denoting the stimulus condition). The

boundaries are not imposed on the accumulating evidence $x(t)$ directly, but on the emergent motor accumulation $y(t)$: We denote the characterizing boundary separation with $a$, with an upper boundary at $\frac{a}{2}$ and a lower boundary at $-\frac{a}{2}$. We assume the starting position of the evidence accumulation $x(0) = x_0$. The leak parameter $\lambda$ determines the lag between the evidence and motor accumulation, and $\beta$ is a scaling parameter. When the motor accumulation in $y(t)$ hits a threshold, the physical response ensues, and a choice is made. In terms of choice-RT data alone, the LIT has one redundant parameter: To make the model estimable, we choose $\beta = \lambda$. Solving for $y(t)$ in Eq 1 results in a time-smoothed version of the original accumulating evidence:

$$y(t) = \lambda \int_{-\infty}^{t} dt' x(t') e^{\lambda(t'-t)}. \tag{2}$$

The time smoothing afforded by the motor accumulation effectively increases the signal-to-noise ratio relative to the primary evidence accumulation signal and reduces accidental noise-driven threshold crossings. Unlike traditional instances of the DDM in which the momentary state of the accumulated evidence itself is being evaluated, here, decisions are based on a time-decaying weighted memory of all the evidence accumulator's past values (hence the term "Leaky Integrating Threshold"). This introduces a lag between the primary evidence accumulation and motor accumulation processes. The leak parameter $\lambda$ determines the extent of the temporal smoothing and the duration of the corresponding lag. When $\lambda$ approaches infinity, the LIT has no memory or lag and behaves as a normal threshold on the primary accumulated evidence. Assuming prestimulus evidence accumulation is $x(t < 0) = x_0$ and motor accumulation $y(t)$ has had enough time to adapt, the motor accumulation starting position becomes $y(t) = x_0$.

## Leak versus boundary adjustments for controlling choice urgency

To manipulate the urgency of making a response, participants completed 4 blocks of the face-versus-car categorization task either under a Speed or an Accuracy instruction (Fig 1A). To fit the behavioral data, we proceeded with the estimation of the LIT using the prepaid method proposed by Mestdagh and colleagues [40]. In order not to impose any constraints on the shape of the non-decision time distribution, we opt for a D*M fit criterion [41]. For this analysis, Speed and Accuracy data are fitted completely independently, resulting in different parameter values for each instruction. In line with Verdonck and colleagues [30], we found that inverse leak $\lambda^{-1}$ and boundary separation $a$ were significantly smaller in the Speed than in the Accuracy condition looking at their intraindividual differences ($\Delta\lambda^{-1} = -0.19$, $[-0.22, -0.15]$ 99% bootstrap confidence interval; $\Delta a = -0.089$, $[-0.12, -0.061]$). In addition, the relative starting position $zr$ and drift speeds $v_{i = 1-4}$ were indistinguishable ($\Delta zr = 0.016$, $[-0.024, 0.059]$; $\Delta v_1 = -0.044$, $[-0.24, 0.14]$; $\Delta v_2 = -0.27$, $[-0.79, 0.033]$; $\Delta v_3 = -0.085$, $[-0.29, 0.11]$; $\Delta v_4 = 0.19$, $[-0.15, 0.6]$).

While the prepaid method is useful for the estimation of parameters of models without a readily accessible likelihood function, it does not allow a direct comparison between models. For model comparison, we use an approximation of the LIT, as has been introduced in [30] and further developed in **Materials and methods** below. This approximation is a reparameterized version of a standard diffusion model, in which the original LIT leak parameter translates to a change in the boundary separation, starting position and non-decision time (Eqs 5, 4 and 15). This allows us to construct an overarching model in which multiple mechanisms of controlling for choice urgency can be compared (Eq 16). Specifically, we compare the LIT-based mechanism (balancing the signal-to-noise ratio of a secondary motor accumulation to bound by controlling its leak parameter) with the 2 most common alternatives, involving changing a

single-stage evidence accumulation threshold: either by changing its overall value (boundary separation [8,35]) or by allowing different rates of boundary collapse (collapse rate [33]).

Because the dynamics of this model set are described by a simple one-dimensional diffusion process, we can use a grid method as in Voss and Voss [42] to calculate the associated likelihood function. Again, using D*M as a fit criterion, we estimate 3 implementations, each of them relying on a different parameter to explain the differences between urgency conditions in the data: inverse leak $\lambda^{-1}$, boundary separation ($a$), or collapse rate ($c$). We emphasize that changes in the inverse leak were reparameterized as equivalent changes in the DDM parameters, such that all of the resulting models have the same number of free parameters, and we can use their fit to the data as a way to arbitrate between the mechanisms (Fig 2A–2D). For the great majority of participants, the LIT-based inverse leak parameter ($\lambda^{-1}$) is more successful at explaining differences between urgency conditions than both simple boundary separation $a$ (72%, [51%, 88%] 99% bootstrap confidence interval) and collapse rate $c$ (81%, [63%, 93%]).

In addition, we fit a standard DDM and the LIT (without reparameterising) using the more common simulation-based approach to estimate the negative log-likelihood of response-time quantiles (see Materials and methods). The predicted choice-RT distributions of these models are quite similar (Fig 2E). In this case, the LIT has an additional parameter compared to the DDM and yet not only showed superior negative log-likelihood ($\Sigma NLL_{LIT} = 6.430e^4$; $\Sigma NLL_{DDM} = 6.448e^4$) but also survived the penalty for the additional parameter afforded by the Bayesian information criterion (BIC) ($\Sigma BIC_{LIT} = 1.311e^5$; $\Sigma NLL_{DDM} = 1.312e^5$; on average, LIT BIC was

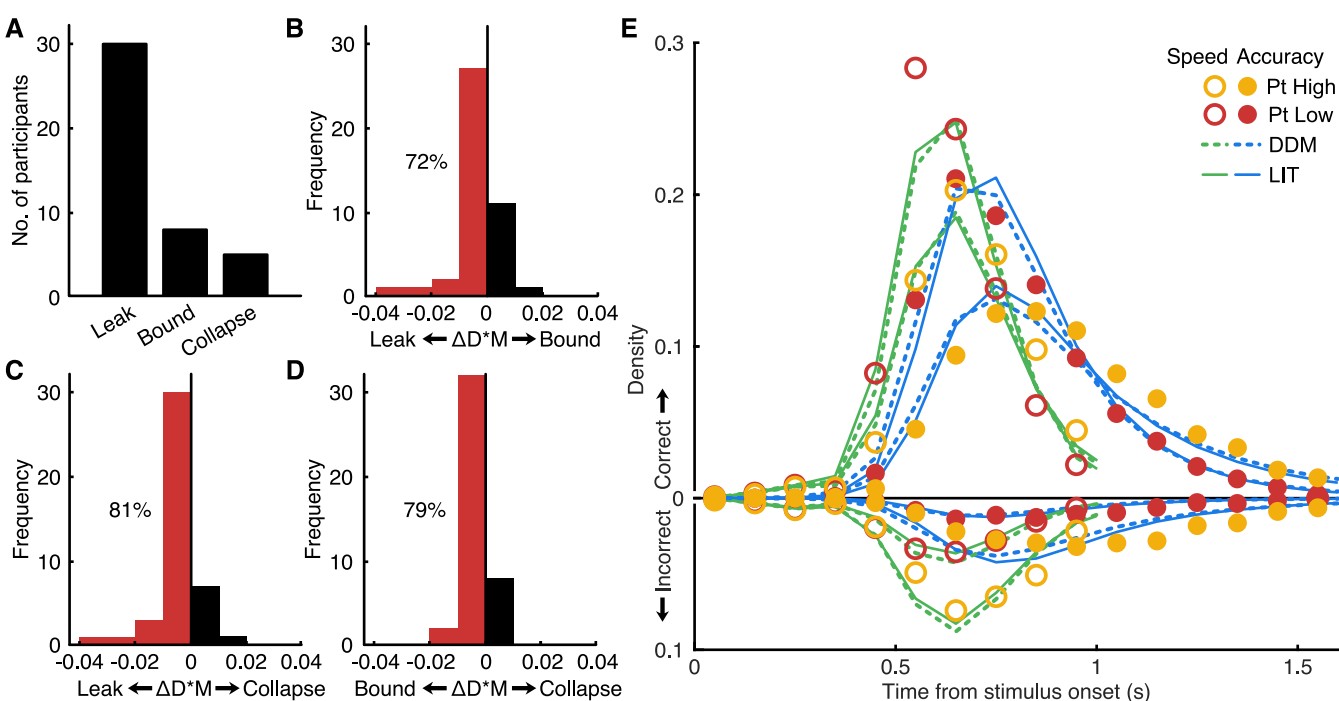

**Fig 2. Comparison of 3 competing mechanisms for controlling choice urgency.** (A) The number of participants for whom each mechanism was best based on the D*M fit criterion (with a total of 43). (B) Frequency histogram of the difference in D*M, where negative values favor leak over bound mechanisms. (C) The same as (B) comparing leak and boundary collapse. (D) The same as (B) comparing boundary separation and boundary collapse, with negative values indicating evidence in favor of the boundary separation mechanism. The constraints of model parameters for (B–D) are summarized in **Table A in** S1 Text. (E) Density of correct and incorrect (plotted downward from 0) responses in equisized bins of response times summarized across participants (markers; yellow for high sensory evidence, red for low; open markers show the speed condition) and prediction from models (based on simulation; DDM with boundary separation mechanism in dashed and LIT in solid lines; green lines show the speed condition).

1.38 less than the BIC for the DDM). Thus, these 2 model fitting approaches suggest similar conclusions: that the leak mechanism of the LIT provides a better computational description of SAT control in the behavioral data.

## Electrophysiological predictions of LIT versus standard DDM

To connect any high-level model of evidence accumulation to EEG data, assumptions have to be made about how exactly the theoretical signal can be expected to manifest at the level of macroscale brain dynamics. In accordance with previous work [11,43], we assume that accumulation activity on the scalp (for instance, gradual build-up of the EEG signal) will be observed as having the same sign for both choice alternatives. To accommodate this in a framework of one-dimensional evidence accumulation, the theoretical evidence signal can be recast as an absolute deviation of a prestimulus baseline.

Moreover, as EEG data could reflect a superposition of different processes, we cannot assume that a clean representation of the evidence accumulation signal can be found at any particular location or for any combination of sensors. Regardless of the unmixing technique used [44], the evidence accumulation signal extracted from the EEG data might contain activity from other processes. If we start from the premise that a meaningful model of decision-making will at least include processes that explain some of the differences in choice-RT between trials, a valid comparison between model-derived and actual EEG accumulation signals could consist of looking at intertrial correlations of momentary signal values with their respective final RTs, instead of comparing the raw accumulation signal profiles themselves.

This approach has the important advantage that EEG activity that is shared across trials but does not influence final choice-RT outcomes (and hence may be unrelated to the process of interest) is filtered out. More specifically, correlations of momentary signal values with their corresponding final RTs are not affected by an "unspecific" shift in the signal values, as long as it is present in all trials. In other words, adding a constant signal to all trials—although very much altering the resulting mean accumulation signal—would not impact the correlation of momentary accumulation values and RTs. These accumulation "signal-RT" correlations can be calculated in a response-locked fashion, for different points in time leading up to the eventual response.

The ensuing signal-RT correlation curves have an amplitude that can be interpreted as the extent to which the eventual trial RTs can be predicted by their preceding accumulation signal at a given point in time. The sign of the signal-RT correlation indicates whether higher accumulation signals lead to slower RTs or the other way around. In trials where the accumulation signals of slow and fast trials are comparable in value, the signal-RT correlation is zero. On the whole, signal-RT correlation curves are driven by how trials with different response times (from slow to fast) evolve compared to each other, and not how their mean value evolves compared to the rest of the brain activity. Signal-RT correlation curves can therefore be seen as a robust metric to compare the internal dynamics of theoretical evidence and motor accumulation signals with what is observed in the EEG signals.

To assess these properties in both the standard DDM and its LIT extension, we simulated all accumulation signals and their corresponding signal-RT correlations. In Fig 3A, participant-averaged DDM evidence accumulation signals for the 2 levels of sensory evidence are shown locked to the crossing of the decision threshold ($t = 0$). At all times before the crossing, the low sensory evidence signal is higher than the high sensory evidence signal (i.e., the accumulation signals never cross before the final response). The difference between the 2 signals reduces to zero only at $t = -\infty$ and at the threshold crossing ($t = 0$). The latter is consistent with the idea of a stimulus invariant information threshold, whereby all evidence

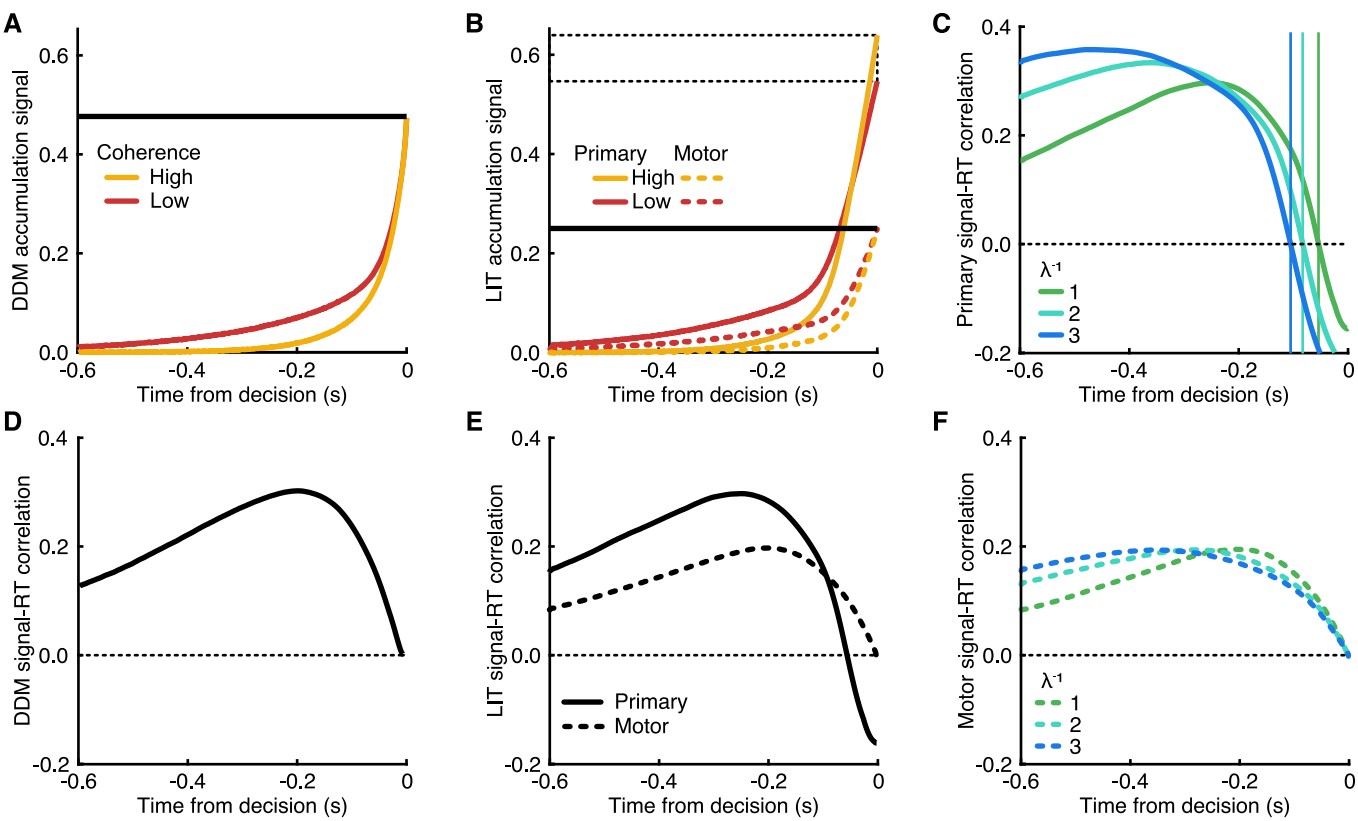

**Fig 3. Model predictions. (A)** Evidence accumulation signal of the DDM locked to decision time, averaged over 100,000 simulations with independent noise. The yellow line shows high sensory evidence strength and red low; the horizontal black line marks the boundary. **(B)** The LIT model proposes 2 co-temporal accumulation signals (primary in solid and motor in dashed lines). The boundary (thick black line) is implemented on the motor accumulation signal, leaving the primary unbounded, meaning that the primary evidence accumulated up to the time of the decision is modulated by the sensory evidence strength (greater for high than low; horizontal dashed lines). **(C)** Under the LIT framework, the correlation between the primary evidence accumulation signals and response times switches from positive to negative prior to the decision, and the timing of this zero-correlation (thin vertical lines) is related to the motor accumulator leak (different colors). **(D)** In contrast, for the DDM framework, the correlation reaches zero at the time of the decision. With additional non-decision time prior to entering the response, the signal-RT correlation would approach zero earlier relative to the response but would never become negative. **(E)** Comparison of signal-RT correlations for the primary (solid) and motor (dashed) accumulators of the LIT, where the motor signal-RT correlation behaves similarly to the DDM signal-RT correlation. **(F)** The motor signal-RT correlation of the LIT reaches zero at the time of the decision, unmodulated by leak (colors). Instead, the timing of the peak correlation (inflexion from increasing to decreasing) changes with leak. Parameters for these simulations are summarized in **Table B in S1 Text**.

accumulation signals at $t = 0$ have the same value. Correspondingly, the evidence signal-RT correlation curve remains positive during this time interval (higher values lead to slower RTs; Fig 3D) but reduces to zero at $t = -\infty$ and $t = 0$.

This observation stands in stark contrast with the behavior of the LIT (Fig 3B) and recent EEG work demonstrating a stimulus-depended signal modulation at the time of response [11,14,45]. The mean LIT evidence accumulation signal for low sensory evidence dominates the signal for high sensory evidence at the early stages, but later, the 2 signals cross over as the signal for high sensory evidence begins to dominate. At the motor threshold ($t = 0$), the signals for the 2 levels of sensory evidence do not coincide, demonstrating a clear deviation from the idea of a stimulus invariant information threshold on the level of the evidence accumulation. This pattern also manifests in the signal-RT correlation (Fig 3C) curve via a sign switch at the cross-over point between the evidence accumulation signals, ending up negative at the time of response. Importantly, the LIT leak parameter impacts the profile of the accumulation-RT

correlation curves (Fig 3C). Specifically, higher values of inverse leak $\lambda^{-1}$ (i.e., more time smoothing applied going from evidence accumulation to motor accumulation) result in an increased lag between motor and evidence accumulation signals. This lag can be quantified by the response-locked time at which the signal-RT correlation curve crosses zero (Fig 3C; vertical lines).

For the LIT, the final decision criterion is not acting on the evidence accumulation itself, but rather on a secondary motor accumulation (Fig 3B; dashed lines) that is driven by the primary evidence accumulation. The corresponding motor accumulation signal-RT correlation curve equals zero at the response (Fig 3E), consistent with the fixed criterion imposed on this accumulator, and this is independent of the inverse leak (Fig 3F), which instead modulates the timing of the inflexion (from increasing to decreasing correlation) relative to the decision (peak signal-RT correlation).

By directly modeling motor processes, the LIT assumes the response (button press) is executed with only a small and systematic delay relative to the time the motor threshold is reached. However, for the standard DDM, the non-decision time is assumed to include additional time for the choice to be communicated to the motor cortex, after which the button press is planned and executed. If evidence accumulation were to continue during this time with the same drift rate, and the time-course of evidence accumulation was locked to the time of the response, rather than the decision, the signal-RT correlation could become negative so long as there is not substantial variability in the non-decision time across trials. The timing of this zero-crossing is relative to the duration of non-decision time. Under the DDM framework, the signal-RT correlation is dependent on the sensory evidence strength but not on the decision boundary separation, meaning that this model predicts no substantial difference in signal-RT correlations between speed and accuracy conditions.

In the next section, we compare these theoretical signal-RT curves with the corresponding curves derived from EEG accumulation signals to offer neurobiological validation for the proposed LIT framework. Specifically, the LIT predicts that the primary evidence accumulation signal-RT correlation will cross zero, and the timing of this zero-crossing will be earlier in the accuracy condition than the speed condition (as it is dependent on the leak). In contrast, the DDM predicts signal-RT correlations that do not substantially differ between the speed and accuracy conditions.

## EEG signatures of LIT-like evidence accumulation

To offer neurobiological validation for the LIT, we first aimed to identify candidate accumulator signals from which to derive the required signal-RT curves. Specifically, we used a single-trial multivariate classifier [38,46,47] designed to estimate linear spatial weightings of the EEG sensors discriminating between low and high levels of sensory evidence (collapsing across Speed and Accuracy trials) as was done in [11] (see Materials and methods). Applying the estimated electrode weights to single-trial data produced a measurement of the discriminating component amplitudes, which we treat as a neural surrogate of the relevant decision variable being integrated. We performed this analysis on response-locked data and separately for each participant, with discrimination performance (area under the receiver operator characteristic (ROC)) shown in Fig 4A.

Importantly, this approach offered an initial opportunity to arbitrate between the predictions of the traditional DDM and the LIT. The traditional DDM assumes a common threshold for the primary evidence accumulator for both low and high evidence trials. This, in turn, should lead to low discrimination performance near the response with systematic improvements in classification accuracy further back in time as the traces associated with the different

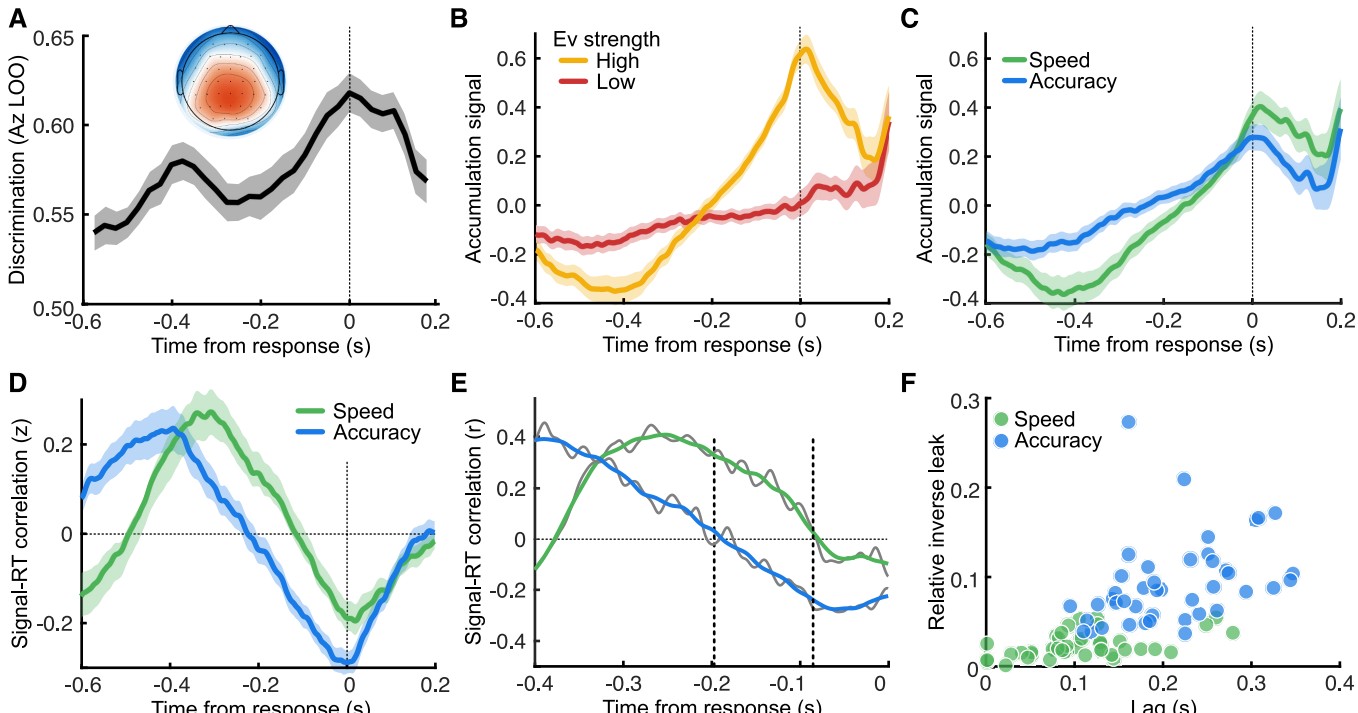

**Fig 4. Evidence accumulation signals in the EEG.** (**A**) Performance (area under the ROC, Az, based on a leave one out analysis) of the multivariate classifier discriminating high from low sensory evidence trials from EEG activity locked to the response. The insert shows the topography of the forward model at the time of the response, where red indicates a greater influence on discrimination, and blue less influence. (**B**) The EEG-predicted evidence accumulation signals by applying the sensor weights from the multivariate classifier at response time, averaged over high (yellow) and low (red) sensory evidence strength trials. Shaded regions show parametric 95% confidence intervals. (**C**) The same as (**B**) but averaged over speed (green) compared to accuracy (blue) trials. (**D**) Average Fisher-transformed (z) Pearson correlation between the EEG-predicted evidence accumulation signals and response times for speed (green) and accuracy (blue) conditions. Shaded regions show parametric 95% between-subject confidence intervals. (**E**) The data for (**D**) from a single participant, showing the moving average of 50 ms used to estimate the zero-crossing. (**F**) Lag (seconds prior to response) of the signal-RT correlation zero-crossing by inverse lambda (relative to bound separation) for each participant (separate markers) in the speed (green) and accuracy (blue) conditions.

levels of sensory evidence begin to deviate (Fig 3A). In contrast, the LIT predicts that classification accuracy, in so far as it is driven by the evidence accumulation signal, should be highest near the response, drop gradually moving back in time as the traces of the different levels of sensory evidence cross and pick up again as they begin to deviate in the opposite direction (Fig 3B). In other words, we should observe 2 separate discrimination peaks in the period leading up to the response.

We note that the LIT motor accumulation signal (i.e., secondary accumulation) may also contribute to the classification accuracy further away from the response but, because it converges to a common threshold, cannot explain any potential above chance classification near the response itself. We observed 2 local discrimination maxima, one near the response itself and one approximately 400 ms prior the response, on average (Fig 4A). This classification performance profile fits squarely within the LIT predictions, offering initial support for this framework. The spatial topography (forward model; see Materials and methods) of the discriminating activity at the time of response (insert of Fig 4A) is consistent with the spatial distribution of centroparietal EEG signals implicated previously in the process of evidence accumulation [11,43,48].

We subsequently applied each participant's spatial weights obtained in the 50-ms time-window centered around the response (i.e., window of best overall discrimination performance)

on the entire response-locked time window to obtain individual trial-by-trial evidence accumulation signals. This approach can be thought of as projecting the data through the same neural generators responsible for the main discriminating activity and was designed to obtain robust estimates of the full temporal profile of this activity beyond the point of maximum discrimination [11,49].

An initial qualitative assessment of the resulting temporal profiles is in line with what is predicted by the LIT (Fig 3B), whereby there is no common decision boundary for low and high sensory evidence trials close to the response. On the contrary, it is the moment where the difference between the 2 signals is most pronounced (Fig 4B). Additionally, the signals appear to cross over some time before the response (circa 200 ms prior to the response), creating 2 regions where they are separable: an early one where the high sensory evidence signal is lower than the low sensory evidence signal, and a later one near the response itself where the effect is reversed. Similarly, stratifying trials according to the SAT manipulation suggests there are no boundary differences across the speed and accuracy evidence accumulation signals near the response, on average Fig 4C).

Importantly, these temporally resolved evidence accumulation signals could be examined formally in the framework of signal-RT correlation curves and be directly compared to the theoretical predictions of the LIT. This approach is generally more robust against potentially interfering trends from unspecific neural processes that mix into the EEG signal and are not coupled to RT. In line with the LIT model predictions, both the Speed and Accuracy evidence accumulation signals show trial-by-trial correlations with RT that are negative at response time and become increasingly less negative going back in time until they cross the zero-correlation line while they continue to become increasingly positive (i.e., compare Figs 3C and 4D).

Next, we define motor-lag as the time between the zero-crossing of the evidence accumulation-RT correlation curve and the actual response, since we expect the motor accumulation signal to reach zero correlation with RT very close to the response itself (this is implied by a fixed threshold, where the signal values of all trials are equal). If we assume the inverse leak parameter $\lambda^{-1}$ in the LIT is largely responsible for SAT control (as shown in Verdonck and colleagues [30] and replicated here—the inverse leak is larger for accuracy than speed in 42 out of 43 participants), then the LIT makes a very specific prediction: The motor-lag for Speed should be markedly smaller than the one for Accuracy (Fig 3C). We indeed see this pattern emerge in our population EEG data (Fig 4D) as well as in individual participants (Fig 4E). This pattern is inconsistent with the DDM, which predicts no substantial difference in signal-RT correlations depending on boundary separation (the primary mechanism for SAT control under the DDM framework).

Finally, we also find a clear correlation between $\lambda^{-1}$ (obtained solely from behavioral choice-RT data) and the individual motor-lags (obtained purely from the EEG data), such that participants with higher inverse leak $\lambda^{-1}$ (relative to boundary separation) also exhibit a longer motor-lag (a Spearman correlation of 0.73, $p = 1.38 \times 10^{-15}$; Fig 4F). This is consistent with the notion that higher values of $\lambda^{-1}$ lead to more temporal smoothing for the motor accumulation, thus increasing its signal-to-noise ratio, and decreasing the chance of responding incorrectly, while at the same time leading to an increase in motor-lag, resulting in slower responses.

## EEG signatures of LIT-like motor accumulation

Having established the presence of evidence accumulation signals in the EEG consistent with the LIT, we then tested the extent to which the relevant motor accumulation signals could also be captured at the macroscopic level of scalp potentials. Theoretically, the motor accumulation signal can be derived from an already established evidence accumulation signal. Specifically,

Eq 1 suggests there should be a linear relation between the change of the motor accumulation signal (i.e., slope) and the value of the evidence accumulation signal, at every time $t$ the LIT is operational. We applied this theoretical framework to derive motor accumulation signals directly from the EEG signals, which we then used as a benchmark for further validating the LIT framework.

More concretely, we first calculated, for each trial separately, the local raw signal slopes for all EEG channels (in windows of 50 ms in duration) at different times $t$. As detailed in **Materials and methods**, we calculated these slope snapshots (each consisting of 64 slope values—one for each EEG channel) every 25 ms, with window centers ranging from 150 ms to 50 ms before the response. We expect the build-up of the proposed motor accumulation signals to be most pronounced in this interval due to increases in corticospinal excitability in anticipation of the eventual motor response [50]. Similarly, this is the interval over which evidence accumulation signals peaked (see Fig 4B), and, hence, the leaky integrating response of the motor accumulator should be most pronounced during this period.

From the slope snapshots (number of trials × number of time windows), we then estimate, for every participant separately, the optimal sensor weights for maximizing the correlation between the momentary slopes in the raw sensor signals and the concurrent values of the evidence accumulation signals we derived in the previous section (see Materials and methods). According to the LIT, the momentary slope of a motor accumulation signal should indeed be proportionate to the corresponding momentary value of evidence accumulation (Eq 1). By finding the weights that make this feature most pronounced, we construct a motor accumulation signal that is maximally compatible with one of the fundamental assumptions of LIT dynamics. We calculated these weights using only the trials of the high sensory evidence condition because these trials have the most pronounced evidence accumulation signals and will be better at teasing out the corresponding motor accumulation slopes.

Next, we applied each participant's resulting sensor weights on the entire response-locked time window to obtain robust estimates of the full temporal profile of individual trial-by-trial motor accumulation signals (Fig 5A and 5B). Furthermore, we estimated the spatial topography of the proposed motor accumulating signal by calculating its correlation with each individual sensor signal, using data from all trials (Fig 5C) as well as separately for left and right choice trials (Fig 5E) in order to harvest the well-known contralateral motor bias [22, 51]. Importantly, these topographies are calculated purely based on LIT model assumptions and EEG-derived evidence accumulation signals. As such, they are not informed about the laterality of the motor response, which we will use as further validation for the presence of motor accumulation signals.

The temporal profile of the proposed motor accumulation signal exhibits a gradual build-up of activity, which peaks at the time of response (Fig 5A and 5B). Given the high uncertainties (shaded regions) of the raw signal estimates, it is difficult to judge differences between low and high sensory evidence trials (Fig 5A) or Speed and Accuracy trials (Fig 5B). Additionally, these estimates could be contaminated by signals that show differences between conditions but have no relation to the actual motor accumulation process. Once again, we exploit the inter-trial signal-RT correlation curves to circumvent some of these potential contaminants to validate the specific temporal features of the LIT motor accumulation signal.

Comparing the trial-by-trial signal-RT correlation curves between Speed and Accuracy conditions (Fig 5C), we observed strong agreement with the LIT predictions (Fig 3C). Specifically, the Speed trials (high integration leak) have their maximum accumulation-RT correlation closer to the response compared to Accuracy trials, and both conditions end up close to a zero accumulation-RT correlation around the response. These observations further reinforce

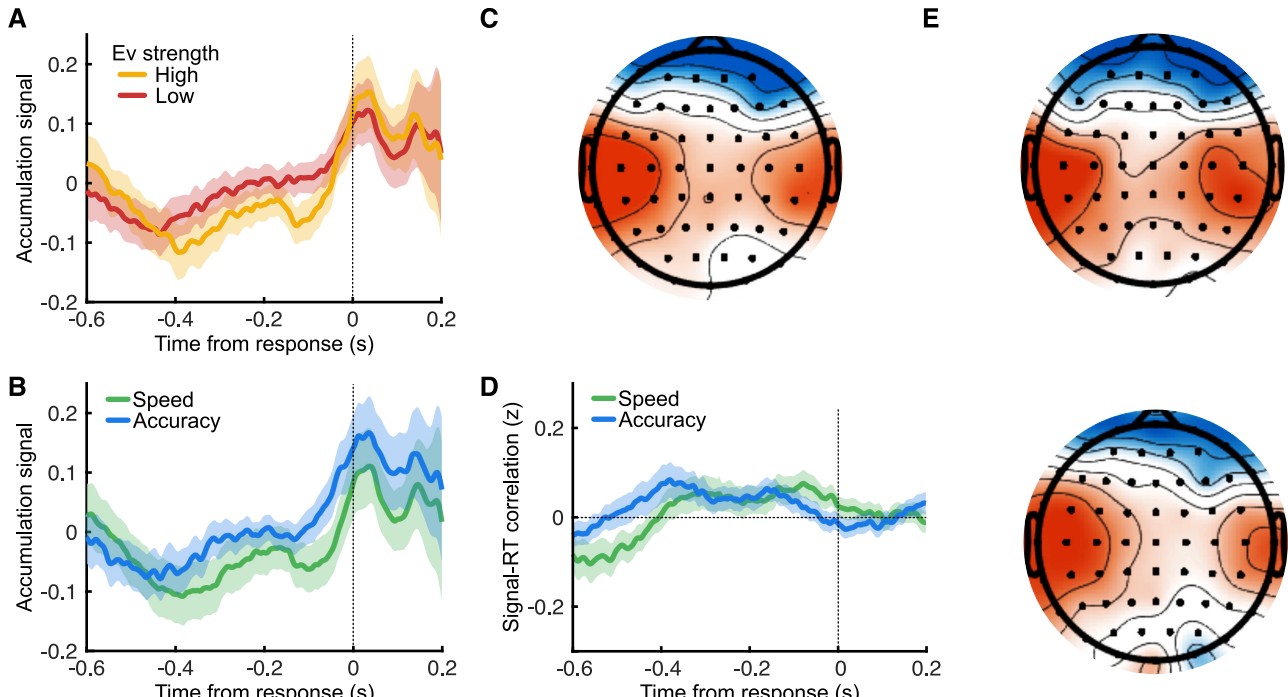

**Fig 5. Motor accumulation signals in the EEG. (A)** LIT-informed motor accumulation signals for low (red) and high (yellow) sensory evidence trials, averaged across participants. Shaded regions show parametric 95% confidence intervals. **(B)** Same as **(A)** but averaging speed (green) and accuracy (blue) trials. **(B)** Topography of the correlation between EEG sensors and the LIT-informed EEG motor accumulation signals at the time of the response. Red shows greater correlation and blue weaker correlation. **(D)** Average Fisher-transformed (z) Pearson correlation between the EEG-predicted motor evidence accumulation signals and response times for speed (green) and accuracy (blue) conditions. Shaded regions show parametric 95% between-subject confidence intervals. **(E)** Calculating the topography separately for trials where the response was made with the left hand shows a greater contralateral motor influence (top). The opposite effect is visible for trials with right-hand responses (bottom). Same color scale as in **(C)**.

the notion of a common motor accumulation boundary and a SAT implementation via leak adjustments of the secondary, motor accumulator.

Finally, we used the spatial topographies of the process of motor accumulation to offer further validation that these signals are indeed emerging from (pre)motor brain structures over the motor strip. Motor preparatory activity has a prototypical scalp distribution with activation clusters over centrolateral sensors and a pronounced contralateral response bias. Here, we mapped the 2 choices on separate hands, and we would therefore expect scalp topographies associated with motor accumulation signals to peak contralaterally to the motor effector used to indicate each choice (i.e., over left [right] centrolateral sensors for right [left] button presses). The topographies we derived for our motor accumulation signals are perfectly consistent with the aforementioned scalp profiles (Fig 5E), offering further validation of the LIT.

## Discussion

In this work, we propose a new computational framework for operationalizing sensorimotor decisions [30] and offer neurobiological validation of the proposed mechanism using human EEG data. Specifically, in the context of traditional evidence accumulation models of choice-RT, we introduce a secondary motor-related leaky integration process (LIT), which receives already stochastically accumulated evidence from a primary process of evidence integration. This doubly integrated signal manifests as a lagged and smoothed version of the original

evidence accumulation signal and controls the eventual choice as it crosses its own threshold, while allowing the primary accumulation process to remain unbounded.

This framework was borne out of necessity to reconcile discrepancies between traditional diffusion models and recent experimental observations in the literature. First, there has been evidence pointing to stimulus-depended values of accumulated evidence at the time of choice ([9–14]; contrary to the traditional view of a common decision boundary). Second, there has been support for a more active role of (pre)motor areas during sensorimotor decisions [15–17,22], with inactivation of such regions leading to gross behavioral impairments [18–21,52].

An emerging trend in the literature focuses on the potential utility of single-accumulator models with collapsing boundaries [33,34] (though this remains contentious [31,32]). This class of models incorporates a continuously encroaching deadline, which dynamically changes the evidence criterion to account for the urgency to make a response. Here, we offer an alternative formulation that additionally considers the likely role of the motor system, while at the same time offering a more flexible account of decision-making with potentially wider implications (see below).

In the case of collapsing boundaries, the model adjusts the end criterion by monitoring the noisy accumulated evidence at discrete points in time without "remembering" past observations (i.e., boundaries are "memoryless"). In the LIT, instead of actively changing the end criterion of the primary accumulator, a second (motor) integration is used to make the signal less noisy to avoid unwanted accidental crossings. In this framework, the motor boundary has a leaky memory, which enables it to consider past evidence rather than just monitor the integrated evidence for an immediate crossing.

Here, we first show computationally that the LIT not only offers a better fit to behavior compared to traditional single-accumulator diffusion models but also provides a vitally different perspective on how choice urgency alters decision dynamics. Contrary to the contemporary view that boundary adjustments (dynamic or otherwise) control the speed versus the accuracy of a decision [8,35], here, we demonstrate that it is a change in the leak of the motor accumulator that instantiates this trade-off. These findings also help reconcile recent neural and computational reports showing little evidence of collapsing decision boundaries as the primary control mechanism for SAT manipulations [36,37,53–55] but rather highlight the involvement of changes in the acceleration of evidence toward the bound [56].

Next, we offer neurobiological validation of the proposed LIT dynamics by leveraging high temporal precision human electrophysiological signals against which to compare the relevant modeling predictions. We capitalize on the fundamentally different accumulation signal predictions of the LIT compared to traditional diffusion models to arbitrate between the competing theoretical constructs. Specifically, we focus on how EEG accumulation signals manifest spatially on the scalp, how they behave as the decision unfolds as a function of stimulus-evidence, as well as how they dynamically relate to the eventual choice-RTs.

In line with the LIT predictions, we identify separate signatures of decision-related accumulation with 2 distinct spatiotemporal profiles. Specifically, an initial evidence accumulation signal with a centroparietal activation profile [11,43,57,58] is trailed by a secondary accumulation signal emerging over the motor strip, with higher activity contralateral to the motor effector used to indicate the decision, consistent with the well-known contralateral motor bias [22,51]. Importantly, we also show that the temporal lag between the 2 EEG accumulation signals across participants scales systematically with the model-derived rate of information leakage in the motor accumulator. To further alleviate potential confounds related to the quality of the unmixing of the 2 EEG accumulation signals, we also look at the intertrial (i.e., point-wise) correlations of these signals with their final RT outcomes and are able to reliably reproduce their theoretically derived counterparts.

To our knowledge, this is the first formal, fully estimable model in which a separate (secondary) motor accumulation process becomes part of the causal chain of events, by receiving stochastically accumulated decision evidence and ultimately taking control over the primary accumulation process to drive the final decision. The smoothing of the evidence accumulation process happening at the LIT stage makes the effective noise of the final accumulator autocorrelated, which is a fundamental difference from most choice-RT models: In previous models with evidence gating, the evidence becomes effectively amplified (for urgency-gating [27–29]), or selectively inhibited (in the case of the gated accumulator model [25,26]), but there is no temporal smoothing. Without temporal smoothing, changes in the amplification/inhibition of evidence over time can be computationally approximated by changes in the temporal profile of the decision boundary [59]. In contrast, double integration provides clearly distinguishable computational predictions promoted by the temporal lag involved in smoothing. We believe this mechanistic proposition is the main novelty of this work, which aligns with insights derived from neurophysiological observations in animals [60–62], and could ultimately have wider implications in how sensorimotor decisions are being formed. For example, the LIT could be relevant in scenarios involving integration of non-stationary evidence with varying degrees of temporal uncertainty [63] by controlling the integration time constant and adjusting the leakage of information being used by the motor accumulator.

An additional advantage of the LIT over previous work is that choice is determined by the motor accumulator, which allows the primary accumulator to remain unbounded and continue to accumulate past the decision. This, in turn, could potentially inform secondary decisions involving additional post-decisional deliberation ("change-of-mind" decisions, post-decision metacognitive appraisal, etc. [64–66]) or even influence information processing on subsequent choices (i.e., introduce serial dependencies across trials [67,68]).

For example, consider the case of change-of-mind decisions or double responses [65,69]. Because the LIT is a separate motor accumulator, resetting it after a motor action does not reset the primary evidence accumulator feeding into it. Immediately after (or even during) this resetting, the motor accumulator could resume accumulating, swiftly picking up on new evidence accumulated by the primary accumulator in the interim. In turn, this second wave of motor accumulation can enable the selection of a different response in cases where the new evidence now points toward an alternative choice.

Overall, the LIT has important neurobiological implications as it highlights the need to differentiate between 2 interrelated but largely separate accumulation processes that are likely to take place in different brain networks [70] or even heterogenous neural populations [71]. While the former could be independent of sensory and response modality [15,39,72,73], the latter would emerge from structures controlling the specific motor effectors involved in implementing the decision [15,22,74], consistent with an embodied cognition model [21].

Moreover, we speculate that the all-important interplay between the 2 processes could be implemented via cortico-basal ganglia circuits controlling the proposed rate of information leakage during motor accumulation. In fact, it is likely that the relevant structures would be comparable to those previously proposed to control the urgency to make a choice by adjusting decision boundaries [35,75–77], including work involving oculomotor decisions [20,21,53,78].

In conclusion, our work provides a novel and biologically plausible alternative to the traditional single evidence accumulation models of choice-RT. Correspondingly, our findings could help revise existing theories of sensorimotor decision-making by imposing new mechanistic constraints, while at the same time offering a new benchmark against which the neural systems underlying such decisions can be interrogated.

## Materials and methods

### Ethics statement

The study was approved by the College of Science and Engineering Ethics Committee at the University of Glasgow (CSE01353), and informed consent was obtained from all participants. The investigation was conducted according to the principles expressed in the 2008 Declaration of Helsinki.

### Participants

Forty-three right-handed volunteers (18 males) aged between 20 and 48 years (mean = 26 years) participated in the experiment. All participants had normal or corrected-to-normal vision and reported no history of neurological problems.

### Stimuli

We used a set of 40 face and 40 car grayscale images (size 500 × 500 pixels, 8-bits/pixel). Face images were obtained from the Max-Planck Institute of Biological Cybernetics face database Germany [79]. Images of cars were sourced from the web and placed on a uniform gray background. We equalized spatial frequency, luminance, and contrast across all images. We adjusted the magnitude spectrum of each image to the average magnitude spectrum of all images. The phase spectrum was manipulated to generate noisy images characterized by their percent phase coherence [80]. We used 2 different phase coherence values (25% and 30%) to manipulate the difficulty of the decision. At each difficulty level, we generated multiple frames for each image whereby the spatial distribution of the noise varied across frames, while the overall level of noise remained unchanged. This, in turn, ensured that when we presented these frames in rapid succession, a dynamic stimulus emerged in which relevant parts of the underlying image were revealed sequentially.

### Behavioral paradigm

Participants performed a visual categorization task by discriminating dynamically updating sequences of either face or car images [11,39]. Participants sat at a distance of 75 cm from the presentation screen such that each image subtended approximately 6 × 6˚ of visual angle. Image sequences were presented in a rapid serial visual presentation fashion at a frame rate of 30 frames per second (i.e., 33.3 ms per frame without gaps) using Presentation (Neurobehavioral Systems, Albany, California). Each trial consisted of a single sequence of noisy images from either a face or car stimulus, at one of the 2 possible phase coherence levels. We introduced separate Speed and Accuracy instruction blocks (2 blocks per instruction, the order of which was pseudorandomized across participants). To encourage a SAT, we introduced different response deadlines for the 2 types of blocks (Speed: 1 s, Accuracy: 1.6 s).

Participants indicated their choices on a response device using the index finger of their right and left hands. The mapping between face/car choices and left/right index fingers was counterbalanced across participants. Once participants made a response, the stimulus was removed from the screen and the trial was terminated. An intertrial interval then followed and varied randomly in duration in the range 1 to 1.5 s. Each block consisted of 160 trials (40 trials per stimulus type and phase coherence level). If participants failed to respond before the deadline, a "Too slow!" message appeared on the screen and the trial was marked as a no-choice trial and was excluded from further analysis.

Prior to the main task, we provided training to enable participants to familiarize themselves with the task and allow them time to settle on an appropriate SAT strategy by learning to pace

themselves accordingly for each block instruction. Specifically, they trained separately under each of the Speed and Accuracy instructions by completing a total of 80 training trials. During the main experiment, each of the 4 blocks also contained 10 training trials before the main experimental trials were deployed. Training also ensured that the influence of ongoing learning effects during the main experiments were minimized.

## Prepaid model estimation

To estimate the LIT, a prepaid estimation method for diffusion models of choice-RT was used [40]. First, a prepaid database of time-scaled LIT probability distribution functions was created, covering a broad range of model parameters. Because we can adjust the timescale of these distributions at very low computational cost when comparing to data, the parameter degree of freedom responsible for time scaling was not included in the final prepaid grid. The prepaid grid used a uniform distribution for the relative starting point bias $zr = 0.5 + \frac{x_0}{a}$ between 0.1 and 0.9 and a uniform distribution for the inverse leak parameter $\lambda^{-1}$ between 0 and 4. Conditional on each of these 2D grid points, 100 drift speeds $v$ were chosen to span accuracies between 0.001 and 0.999. The boundary separation parameter $a$ was determined through time scaling and did not have to be included in the prepaid grid. This was the same grid as used in the original LIT paper [41]. Likewise, a D*M objective function was used as a measure of fit [41]. A D*M objective function allows the estimation of a choice-RT model with an additional, distribution-unspecified non-decision time component in so far that this component is shared across (some) conditions. Using D*M, however, means the time scalar estimator $\hat{s}$ of the original prepaid estimation method [40] can no longer be used. Like in the original LIT paper (although not explicitly mentioned in the text), we used a different estimator, which does not depend on the distribution shape of the non-decision component. The appropriate time scalar was estimated by dividing the standard deviation of the observed mean response times $o_c$ of the 4 different stimulus conditions $c$, by its equivalent calculated on the prepaid model distributions $m_c$:

$$\hat{s} = \frac{\sqrt{\sum_{c=1}^{4} \left(o_c - \frac{1}{4}\sum_{i=1}^{4} o_i\right)^2}}{\sqrt{\sum_{c=1}^{4} \left(m_c - \frac{1}{4}\sum_{i=1}^{4} m_i\right)^2}} \tag{3}$$

Because the non-decision distribution is the same for all stimulus conditions, its mean cancels out when taking the standard deviation of the response time means of the different stimulus conditions, making this formula independent of that non-decision distribution. Additionally, because D*M is time translation invariant, there is no need to shift the time-scaled prepaid distribution before calculating the D*M objective function value in that grid point. The efficacy of this particular prepaid estimation implementation has been extensively demonstrated in the original LIT paper [30].

## Model selection

**Approximating the LIT as a reparameterized standard drift diffusion model.** In the seminal LIT paper [30], it was established that any LIT choice-RT distribution can be approximated by a standard DDM distribution, transferring the effect of the LIT-specific leak parameter $\lambda$ to the rest of the parameters. The impact of the LIT leak on the starting position and non-

decision time if analyzed in a standard DDM framework were quantified as follows:

$$T'_{er} = T_{er} + \frac{3}{2\lambda} \tag{4}$$

$$x'_0 = x_0 + \frac{v}{\lambda} \tag{5}$$

with $x_0$, $\lambda$, and $T_{er}$ parameters of the original LIT and $x'_0$ and $T'_{er}$ parameters of the resulting DDM. Of note, Eq 4 presupposes a change in the apparent non-decision time according to the leak (due to the temporal smoothing).

In the following, we also determine the approximate effect of the LIT leak parameter $\lambda$ on the boundary separation $a'$, which was not yet formally quantified. To do this, we first establish that the deterministic integration of a process of weakly colored noise, as described by Hagan and colleagues [81], is equivalent to the LIT as described in Eq 1 applied to a standard DDM model.

Hagan and colleagues [81] describe the following process:

$$\frac{dX(t)}{dt} = \frac{\sigma Z(t)}{\varepsilon} \tag{6}$$

$$\frac{dZ(t)}{dt} = \frac{Z(t)}{\varepsilon^2} + \frac{\sqrt{2}}{\varepsilon}\xi(t) \tag{7}$$

with absorbing boundaries on $X(t)$ at $\frac{a}{2}$ and $-\frac{a}{2}$.

If we take the time derivative of Eq 6, we can substitute $\frac{dZ(t)}{dt}$ with Eq 7:

$$\begin{aligned}
\frac{d^2X(t)}{dt^2} &= \frac{\sigma}{\varepsilon}\frac{dZ(t)}{dt} \\
&= \frac{\sigma}{\varepsilon}\left[-\frac{Z(t)}{\varepsilon^2} + \frac{\sqrt{2}}{\varepsilon}\xi(t)\right] \\
&= -\frac{1}{\varepsilon^2}\frac{dX(t)}{dt} + \frac{\sqrt{2}\sigma}{\varepsilon^3}\xi(t).
\end{aligned} \tag{8}$$

On the other hand, for drift speed $v = 0$, the differential equations for a DDM with a LIT are:

$$\frac{dx(t)}{dt} = \sigma^*\xi(t) \tag{9}$$

$$\frac{dy(t)}{dt} = \lambda(x(t) - y(t)) \tag{10}$$

with absorbing boundaries on $y(t)$ at $\frac{a}{2}$ and $-\frac{a}{2}$. If we take the time derivative of Eq 10, we can

substitute $\frac{dx(t)}{dt}$ with Eq 9:

$$
\begin{aligned}
\frac{d^2y(t)}{dt^2} &= \lambda \left[ \frac{dx(t)}{dt} - \frac{dy(t)}{dt} \right] \\
&= \lambda \left[ \sigma^* \xi(t) - \frac{dy(t)}{dt} \right] \\
&= -\lambda \frac{dy(t)}{dt} + \lambda \sigma^* \xi(t).
\end{aligned}
\tag{11}
$$

Combining Eqs 8 and 11, we can identify the $\varepsilon$ and $\sigma$ parameters of the Hagan and colleagues problem [81] in terms of DDM-LIT parameters. Comparing the $\frac{dX(t)}{dt}$ and $\frac{dy(t)}{dt}$ terms, we get:

$$
\varepsilon = \sqrt{\frac{1}{\lambda}}.
\tag{12}
$$

Comparing the $\xi(t)$ terms, we get:

$$
\sigma = \frac{\sigma^*}{\sqrt{2}}.
\tag{13}
$$

Given the result of Hagan and colleagues on the increase in boundary separation [81]:

$$
a' = a + 2\varepsilon\sigma \left| \zeta\left(\frac{1}{2}\right) \right|
\tag{14}
$$

we get

$$
\begin{aligned}
a' &= a + 2\frac{\sigma^*}{\sqrt{2\lambda}} \left| \zeta\left(\frac{1}{2}\right) \right| \\
&= a + 2.0653 \frac{\sigma^*}{\sqrt{\lambda}}
\end{aligned}
\tag{15}
$$

in the LIT context. Drift speed $v$ was set to zero in this derivation but has no impact on it [81]. As is standard practice for the DDM, we fix $\sigma$ to make the model identifiable. In this paper, we choose $\sigma^* = 1$.

**Alternative mechanisms of urgency control.** Finally, we can formulate a model that combines 3 possible mechanisms for controlling urgency: boundary separation $a$, LIT leak $\lambda$, and, finally, collapse rate $c$:

$$
\frac{dx(t)}{dt} = x_0 + \frac{v}{2\lambda} + vdt + \xi(t)
\tag{16}
$$

with effective boundary separation $a' = a + 2.0653 \frac{\sigma^*}{\sqrt{\lambda}} - ct$ and $T'_{er} = T_{er} + \frac{3}{2\lambda}$. For $c > 0$, this implies a bounded response time distribution, because all responses will occur before time $t = \frac{a'_0}{c}$, when the collapse is complete.

**Comparing different mechanisms of urgency control.** For each participant separately, we simultaneously fit the choice-RT data from the 4 stimulus conditions and both urgency manipulations. We consider 3 different models embedded within the framework established in Eq 16. For all models, parameters are assumed to be constant across conditions, except for the drift speed, which may change between stimulus conditions (S) and one single additional parameter, which may change between urgency conditions (U). The non-decision distribution

is also allowed to differ between urgency conditions: As this is implied by the LIT (Eq 4), fixing non-decision time across conditions would handicap the other models and make this model comparison unfair. As shown in **Table A in** S1 Text, the 3 models under consideration differ only in the parameter that is allowed to change between urgency conditions: The first model assumes the leak parameter ($\lambda$); the second model assumes the basic boundary separation ($a$); and the third model assumes the collapse rate ($c$) to be responsible for any differences between urgency conditions.

The probability density functions are calculated using the grid-based fast-dm approach [42] but recoded for GPU for increased performance. Parameters are estimated using a D*M objective function [41]. The objective function is minimized using the differential evolution approach [82]. As all models under consideration have the same number of parameters, we can directly compare fits for model selection purposes.

In addition to the grid-based search, we obtained parameter estimates for each observer for DDM and LIT models without reparameterization, by minimizing the negative log-likelihood of the data using Bayesian Adaptive direct search [83]. The negative log-likelihood was estimated over 10 quantiles of reaction times using a Monte Carlo simulation procedure with 10,000 samples per stimulus condition [84]. Both models had 4 parameters for drift rate ($v$; one for each stimulus condition) and 2 parameters for non-decision time (the mean and variance of a truncated gaussian), which was kept constant across SAT conditions (as here the DDM is not specifically handicapped). For the DDM, SAT conditions were modeled with 2 separate bound parameters, while for the LIT, there was 1 bound parameter and 2 separate leak parameters. Thus, the DDM had a total of 8 parameters and the LIT, 9. Model comparison was therefore conducted using the BIC.

**Parameters used for prediction simulations.** All simulations are done using a normal Euler–Maruyama algorithm with a dt accuracy of 1 ms. The simulations needed to create Fig 3 use the parameters given in **Table B in** S1 Text for each condition. Additionally, all conditions use a starting position $x_0 = 0$. For each condition, 100,000 trials were run. Finally, before calculating the correlation-RT curves, an amount of normally distributed noise (standard deviation 0.015) was superimposed on the mean signals to simulate measurement noise and residues from other processes. The simulated choice-RT distributions in Fig 2E were generated by simulating the models with the fitted parameters of each participant.

## EEG data acquisition

We collected EEG data inside an electrostatically shielded booth using a 64-channel EEG amplifier system (BrainAmps MR-Plus, Brain Products, Germany) and recorded using Brain Vision Recorder (BVR; Version 1.10, Brain Products, Germany) with a 1-kHz sampling rate and an analog bandpass filter of 0.016 to 250 Hz. The EEG cap consisted of 64 Ag/AgCl passive electrodes (Brain Products, Germany) positioned according to the international 10–20 system of electrode positioning, with chin ground and left mastoid reference electrode. All input impedances were kept below 20 kΩ. Experimental event codes and behavioral responses were also synchronized with the EEG data and collected using the BVR software.

## EEG preprocessing

We preprocessed the EEG signals offline using MATLAB (MathWorks, Natick, Massachusetts). Specifically, we applied a bandpass (fourth-order Butterworth) filter between 0.5 and 40 Hz. The filter was applied non-causally to avoid distortions caused by phase delays (using MATLAB "filtfilt"). We then removed eye-blink artifacts using a principal component analysis (PCA) approach. Specifically, prior to the main experiment, we asked participants to complete

an eye movement calibration task during which they were instructed to blink repeatedly upon the appearance of a fixation cross in the center of the screen, while we collected EEG data. This enabled us to determine linear EEG sensor weightings corresponding to eye blinks using PCA such that these components were projected onto the broadband data from the main task and subtracted out [38,85]. Finally, we re-referenced the EEG data to the average of all channels.

## EEG analysis

**Estimating the evidence accumulation signal.** Starting from the assumption that any meaningful evidence accumulation signal should be able to discriminate between stimulus conditions of high versus low sensory evidence, we estimate it as the linear combination of signal components, $y(t)$, which is best able to discriminate between these conditions:

$$y(t) = \boldsymbol{w}^T x(t) = \sum_{i=1}^{64} w_i x_i(t) \qquad (17)$$

with $\boldsymbol{w}$ the sensor weights (spatial filter) resulting in a maximally discriminating $y(t)$.

We perform this linear discriminant analysis across response-locked time points, by using a sliding window approach. Specifically, we define time windows of 50 ms duration and shift the window center with 25 ms increments in a time interval ranging from 575 ms before to 175 ms after the response. For each sensor, we then time-average the samples within each 50 ms window. Using these time-averaged channel values, the sensor weights that maximally discriminate between trials with different levels of sensory evidence, can be found using the Fisher discriminant [86]:

$$\boldsymbol{w} = \boldsymbol{S}_c(\boldsymbol{m}_2 - \boldsymbol{m}_1) \qquad (18)$$

with

$$\boldsymbol{S}_c = \frac{\boldsymbol{S}_1 - \boldsymbol{S}_2}{2} \qquad (19)$$

where $\boldsymbol{m}_i$ is the estimated mean and $\boldsymbol{S}_i$ is the estimated covariance matrix of the time-averaged channel values of the trials of condition $i = 1, 2$ (i.e., weak and strong sensory evidence trials). We quantify the performance of this classifier for each time window by using the area under an ROC curve [87], referred to as $Az$, with a leave-one-out cross-validation approach [86]. Scalp topographies are calculated by projecting the raw signal of each sensor on the already established evidence accumulation signal vector $\boldsymbol{y}$ ($50 \times 1$), evaluated in the 50-ms window centered around the response:

$$\boldsymbol{a} = \frac{\boldsymbol{X}\boldsymbol{y}}{\boldsymbol{y}^T\boldsymbol{y}} \qquad (20)$$

with $\boldsymbol{X}$ ($64 \times 50$), a matrix composed of 64 channel rows covering 50 ms of signal. Vector $\boldsymbol{a}$ describes the electrical coupling between the individual sensors and the evidence accumulation signal. The EEG-predicted evidence accumulation signals were computed using the spatial filter ($\boldsymbol{w}$) from the window around the response ($t = 0$; though choosing a window prior to the response produces similar results) and were z-scored prior to averaging across trials and participants. Correlations between this signal and response times were conducted with a 50-ms moving average to smooth out noise.

**Estimating the motor accumulation signal.** The fundamental dynamical assumption of the LIT is comprised in differential Eq 1, where it is stated that small changes in the motor accumulation signal are proportionate to the momentary value of the evidence accumulation

signal. Based on this, we would expect, for each time $t$ of each separate trial, the momentary slope of the motor accumulation signal to be linearly proportional to the momentary value of the evidence accumulation signal. This proportionality should result in a correlation between all trial-wise momentary slopes from the motor accumulation signals and their corresponding momentary evidence accumulation signal values. In what follows, we try to find those sensor weights and resulting aggregate motor accumulation signal, which maximizes this correlation in the data.

For each participant separately, we calculate a number of slope snapshots. Each slope snapshot is calculated based on the raw signals of a specific trial and pertains to individual 50-ms time windows within that trial (decreasing this duration gives similar, though more noisy, results). It is a vector containing the slopes (linear regression coefficient, with an intercept term) of the signal at every sensor (here, the number of sensors is 64). For each trial, we calculate a 50-ms window slope snapshot every 25 ms with window centers ranging from 150 ms to 50 ms before the response (though the window centers can be extended further prior to the response without affecting the results). The total number of slope snapshots, which is the product of the number of trials under consideration and the number of time windows for each trial, is defined as $K$. From the slope snapshots $S_{sk}$ ($64 \times K$), we estimate the sensor weights $w_s$ that produce a spatially aggregated slope that is maximally correlated with previously calculated evidence accumulation signals $e_k$ ($1 \times K$), using the following formula:

$$\boldsymbol{w} = \bar{\boldsymbol{e}}\bar{\boldsymbol{S}}^{-1} \tag{21}$$

with $\bar{e}$ defined by

$$\bar{e}_k = e_k - \frac{1}{K}\sum_{i=1}^{K} e_i \tag{22}$$

and $\bar{S}$ defined by

$$\bar{S}_{sk} = S_{sk} - \frac{1}{64}\sum_{i=1}^{64} S_{ik} \tag{23}$$

In this analysis, we only use the high sensory evidence trials, to enhance potential left–right hemisphere differences.

## Supporting information

**S1 Text. Supplementary tables. S1 Table A**. Constraints of model parameters across conditions, for each of the models under consideration (see Fig 2A–2D). The constraints are given per model (row): A dot indicates that this parameter is fixed across all conditions; S indicates that the parameter may change between stimulus conditions; and U indicates that the parameter may change between urgency conditions. **S1 Table B**. Prediction simulation parameters used for Fig 3.
(DOCX)

## Acknowledgments

We thank Bharti Gupta for assistance with data collection.

## Author Contributions

**Conceptualization:** Stijn Verdonck, Marios G. Philiastides.

**Data curation:** Tarryn Balsdon, Marios G. Philiastides.

**Formal analysis:** Tarryn Balsdon, Stijn Verdonck, Tim Loossens, Marios G. Philiastides.

**Funding acquisition:** Stijn Verdonck, Marios G. Philiastides.

**Investigation:** Stijn Verdonck, Marios G. Philiastides.

**Methodology:** Tarryn Balsdon, Stijn Verdonck, Tim Loossens, Marios G. Philiastides.

**Project administration:** Stijn Verdonck, Marios G. Philiastides.

**Resources:** Stijn Verdonck, Tim Loossens, Marios G. Philiastides.

**Software:** Tarryn Balsdon, Stijn Verdonck, Tim Loossens, Marios G. Philiastides.

**Supervision:** Marios G. Philiastides.

**Validation:** Tarryn Balsdon, Stijn Verdonck, Marios G. Philiastides.

**Visualization:** Tarryn Balsdon, Stijn Verdonck, Marios G. Philiastides.

**Writing – original draft:** Stijn Verdonck, Marios G. Philiastides.

**Writing – review & editing:** Tarryn Balsdon, Stijn Verdonck, Tim Loossens, Marios G. Philiastides.

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
