## [Editor Report · Decision Letter 0]

7 Oct 2022

Dear Dr Philiastides, 

Thank you for submitting your manuscript entitled "Motor accumulation as a final arbiter in sensorimotor decision-making: a tale of double integration" for consideration as a Research Article by PLOS Biology.

Your manuscript has now been evaluated by the PLOS Biology editorial staff, as well as by an academic editor with relevant expertise, and I am writing to let you know that we would like to send your submission out for external peer review.

Once your full submission is complete, your paper will undergo a series of checks in preparation for peer review. After your manuscript has passed the checks it will be sent out for review. To provide the metadata for your submission, please Login to Editorial Manager (https://www.editorialmanager.com/pbiology) within two working days, i.e. by Oct 09 2022 11:59PM.

Kind regards,

Kris

Kris Dickson, Ph.D. (she/her)

Neurosciences Senior Editor/Section Manager

PLOS Biology

kdickson@plos.org

---

## [Decision Letter · Decision Letter 1]

7 Nov 2022

Dear Dr Philiastides,

Thank you for your patience while your manuscript "Motor accumulation as a final arbiter in sensorimotor decision-making: a tale of double integration" was peer-reviewed at PLOS Biology. It has now been evaluated by the PLOS Biology editors, an Academic Editor with relevant expertise, and by several independent reviewers. 

In light of the reviews, which you will find at the end of this email, we would like to invite you to revise the work to thoroughly address the reviewers' reports.

Given the extent of revision needed, particularly in light of Reviewer 2's comments regarding the ability to rule out a single-stage model, we cannot make a decision about publication until we have seen the revised manuscript and your response to the reviewers' comments. Your revised manuscript is likely to be sent for further evaluation by all or a subset of the reviewers.

**IMPORTANT - SUBMITTING YOUR REVISION**

*Re-submission Checklist*

*Published Peer Review*

*PLOS Data Policy*

*Blot and Gel Data Policy*

Sincerely,

Kris

Kris Dickson, Ph.D., (she/her)

Neurosciences Senior Editor/Section Manager

PLOS Biology

kdickson@plos.org

REVIEWS:

Reviewer's Responses to Questions

PLOS authors have the option to publish the peer review history of their article (what does this mean?). If published, this will include your full peer review and any attached files.

Reviewer #1: Yes: Jeffrey D Schall

Reviewer #2: No

Reviewer #1: This review was prepared by Jeff Schall. I am familiar with and have benefitted intellectually from the authors' previous D*M work and the Leaky Integrating Threshold (LIT) model. I am very sympathetic to the authors' theoretical stance in this work because, as they know, it aligns so well with numerous observations we and others have made, and it duplicates theoretical stances we have adopted previously. Here, the authors have used their insightful and original LIT model to distinguish EEG signals putatively associated with stimulus categorization and response preparation. The approach is sophisticated. The results are not beyond methodological critique by skeptics, but they are intelligible and entirely consistent with previous observations. The interpretation follows naturally from the results. The manuscript is written clearly enough for diverse scientists. The theoretical perspective and the observations should make a very useful contribution to the literature on model-based decision making of general interest to a broad audience. However, as detailed below, the manuscript can be improved in scholarship to enhance its impact.

(1) The authors' appeal to theoretical novelty in the Introduction and Discussion seeks to reach unnecessarily far. In the Introduction, the authors write, "These recent ﬁndings suggest that neural activity related to motor preparation, starts its buildup before the sensory evidence integration completes, effectively lagging the primary process of evidence accumulation. Consequently, the amount of sensory evidence could have a direct impact on motor planning, which in turn could inﬂuence the eventual choice. This entanglement is not accounted for in current incarnations of decision models nor has it been characterized at the level of neuronal responses." This summary and the conclusion are factually incorrect. Multiple neurophysiological studies have described motor preparation beginning before sensory evidence integration completes. Also, the claim that no "incarnation of decision models" accounts for this is contradicted by the "Gated Accumulator Model" formulated by Tom Palmeri, Gordon Logan, and me with our trainees. Now, in Discussion, the authors do acknowledge the alignment of the LIT structure with numerous preceding neurophysiological investigation. They are careful to distinguish LIT as a "formal, fully estimatable model", and that is so. But the authors could be more generous in acknowledging other model-based frameworks that anticipate their original approach such as the "Gated Accumulator Model" (Purcell et al 2010; 2012). Yes, they cite one of the original publications, but the text in the Introduction obscures the convergence of outlooks, which reinforces their preferred theoretical stance. Also, the text in the Discussion offers a naïve reader no insight into the conceptual relationships between these two bodies of research. Such an explanation would not detract from but would rather amplify the authors' impressive demonstration. 

(2) The authors' treatment of the recent literature on neural mechanisms of SAT can be improved substantially. They conclude that "SAT is likely controlled by changes in the proposed leaky motor integration process rather than the long-standing view of boundary adjustments". This conclusion is anticipated by but does not acknowledge as clearly as it might our original observations about the neurophysiology of SAT (Heitz & Schall 2012; see also Reppert et al. 2018). The authors also write, "These findings also help reconcile recent neural and computational reports showing little evidence of collapsing decision boundaries during SAT manipulations.", citing Heitz and Schall 2012 #49 and Hanks et al. 2014 #50 among others. Unfortunately, this sentence conveys only the least of what we and others have found. We discovered that the neurons identified with evidence accumulation and response preparation in frontal eye field and in the superior colliculus did not vary in 'threshold' as predicted by the standard accumulator models (reviewed by Bogacz et al 2010 #32 in the manuscript). The authors may not have noticed, though, that we also discovered that SAT instructions also profoundly affected the neurons representing the evidence for the decision in this task. When we fed these neural data into the Gated Accumulator Model (Servant et al. 2019 - which they now cite), it captured key measures of performance, but flexible combinations of gate and threshold parameters were necessary to explain differences in SAT strategy across monkeys. Does LIT have enough parameters to do this?

(3) In citing the publications addressing collapsing bounds, the authors do not distinguish the claims and positions of authors on both sides of the question. Some authors would be unhappy to be listed alongside others as if this is a decided question or that it is applicable under all testing conditions. The authors can correct this easily by rephrasing sentences and separating citations. 

The authors may also wish to consider these additional comments:

[A] In Discussion, the authors state that they "…show computationally that the LIT … offers a better fit to behavior compared to traditional single accumulator diffusion models…". The authors should explain how this claim relates to other publications that contrast quantities like BIC. For example, is it not the case that this approach will not scale with either the number of parameters or the number of models?

[B] The analysis pipeline for the EEG data included several choices that a skeptic could query. The manuscript would be improved if the authors demonstrated the robustness of the observations relative to variation of the several steps and parameters used in the analysis.

[C] "estimatable" probably should be "estimable"

***Reviewer #2: Verdonck et al. have recently proposed a dual integration model of perceptual decisions that are reported with a motor action (the Leaky Integrating Threshold or LIT model). Rather than applying a decision threshold to the integrated sensory evidence, as in the case of a typical single-stage integration-to-threshold model like the drift diffusion model (DDM), the integrated evidence is fed into another (leaky) integrator, which ends up being compared to a decision threshold. The authors propose that the mechanism for implementing the speed-accuracy tradeoff is a change in the time constant of the second integration stage.

In this manuscript the authors provide further experimental evidence for the two-stage model based on human behavioral and EEG data from a dynamic face vs. car categorization task.

I find the possibility of a dual-integration model interesting and some of the provided evidence, the observation of spatially segregated EEG signals with an appropriate relationship with each other as well as with the timing of individual decisions, quite convincing. At the same time, I have some questions about some of the choices/statements made by the authors, as further detailed below.

1) The estimation method for obtaining the model parameters (D*M) is based on RT distributions, but we never get to see any RT distributions, neither those of the data, nor those predicted by the considered models. Could you please show some comparisons?

2) One of the characteristic predictions of the LIT model is that a change in the integration time constant of the second stage leads to a change in the timing of the decision that would have to be captured by a change in the nondecision time in the case of a standard DDM. I find it therefore quite surprising that the authors allowed the nondecision time distribution to change with speed/accuracy (urgency condition) in all their model fits, which could potentially undo those kinds of timing changes. Furthermore, as far as I can tell, the resulting estimates of the nondecision time distributions are never reported. Could you please show these (or demonstrate that the LIT model still performs well, when the nondecision time distribution is not allowed to change with speed/accuracy)?

3) Related to this topic, the authors cite a few studies, which have reported that the nondecision time seemed to change with speed/accuracy when applying a DDM to behavioral data. This, however, is not true for all studies. Palmer et al. (2005) is a good example, where decision times were changed over a wide range based on speed/accuracy instructions, but a DDM was able to capture the behavior quite well with only a change in the decision bound and no systematic change in the nondecision time. It should at least be mentioned that there are studies that did not find a change in the nondecision time with speed/accuracy manipulations.

4) In Fig. 3a the authors show accumulated evidence signals from a DDM locked to the decision threshold crossing. It is generally believed that the decision threshold crossing precedes the onset of the motor response and that part of the "nondecision" time that typically is fitted when fitting standard integration-to-threshold models is related to the preparation/execution of the motor response. Model signals that are time-locked to the decision threshold crossing should therefore not directly be compared with neural signals that are time-locked to the onset of the motor response. In fact, models like the DDM do not specify how the accumulated evidence signal should behave after the decision threshold has been crossed, as it is not relevant for the outcome of the decision. It is, however, quite relevant when a comparison between neural activity immediately before the onset of the motor response and model predictions is made. The fact that accumulated evidence reached a fixed level at the time of the threshold crossing, regardless of the strength of the sensory evidence, does not mean that this would still have to be the case at the time of the onset of the motor response. It could, if integration stopped immediately at the time of the threshold crossing without any leak. It would not be expected to be the case, if integration continued after the threshold crossing. (And it is difficult to predict what exactly would happen, if activity started to leak away after the threshold crossing.) I.e., even if a threshold on the accumulated evidence (first and potentially only integration stage) were to terminate the decision, but the integration continued during a (roughly) fixed time interval between the threshold crossing and the onset of the motor action, a crossing of the accumulated evidence signals like the one shown in Fig. 3b would be possible. I would therefore like to see to what extent a single-stage integration model could still account for (some of) the features of the EEG signal that has been linked to accumulated evidence by the authors and that have been interpreted as evidence for the two-stage model (like the sign change of the black solid line in Fig. 3b prior to the onset of the motor response and the corresponding findings in the EEG signal in Fig. 4c and d).

That said, the change in the zero correlation lag with speed/accuracy could probably not easily be explained that way.

5) Hanks et al. (2014) have previously reported invasive recordings from monkey LIP during a perceptual decision task with an urgency manipulation. Are the results that have been reported there (speed stress being mediated by an additive signal before the threshold crossing; not clear across monkeys whether this signal was constant or increasing over time, i.e., added to the output or to the input of an integrator) consistent with the LIT model? If so, would activity in LIP correspond to the first or the second integration stage?

References:

Hanks T, Kiani R, Shadlen MN (2014) A neural mechanism of speed-accuracy tradeoff in macaque area LIP. eLIFE 3:e02260

Palmer J, Huk AC, Shadlen MN (2005) The effect of stimulus strength on the speed and accuracy of a perceptual decision. Journal of Vision 5:376-404

*** Additional feedback from Reviewer 1 (regarding Reviewer 2 comments):

Thoughts on R2 pt 1:

The authors have done this in previous publications but it would be good to include here for completeness.

Thoughts on R2 pt 2:

Very useful point for the authors to address.

Thoughts on R2 pt 3:

A scholarly and reasonably thorough survey of this issue is merited in this manuscript. If the reviewer knows of other studies complementing Palmer 2005, it would be helpful to share them.

Thoughts on R2 pt 4:

The duration of the ‘non-decision’ motor delay is an important detail, and the manuscript would be improved by its consideration. We should note that the duration of such an efferent “point of no return” depends on which body part is moving and in what way. The non-decision motor delay for saccades, for example, is about 10 ms, and that’s all because of the brainstem mechanisms generating the ocular rotation. In the present study, participants indicated their choice via a button press using the index finger of either their right or left hand. The relationship between “decision” and “movement” thresholds has been investigated recently by Mathieu Servant, e.g., (Servant et al JNeurosci 2015 "Using Covert Response Activation to Test Latent Assumptions of Formal Decision-Making Models in Humans"; Servant et al J Exp Psychol Gen 2021 "An integrated theory of deciding and acting").

Thoughts on R2 pt 5:

As noted in my review, the Heitz & Schall (2012) report on neurophysiology of SAT in monkeys reports more happening than the Hanks et al. study. As we are now comparing these studies, we should note that whereas the Heitz study sampled neurons while monkeys alternated between short blocks of Fast and Accurate trials, the Hanks study sampled some neurons when speed was emphasized and other neurons when accuracy was emphasized. In their data, no neuron was sampled in both conditions. This limitation confounds confident interpretations. The authors should also be aware of the Thura & Cisek work on neurophysiology of SAT in a different task (i.e. Thura et al. J Neurosci 2014 "Context-dependent urgency influences speed-accuracy trade-offs in decision-making and movement execution")

---

## [Decision Letter · Decision Letter 2]

8 Jun 2023

Dear Dr Philiastides,

Thank you for your patience while we considered your revised manuscript "Motor accumulation as a final arbiter in sensorimotor decision-making: a tale of double integration" for publication as a Research Article at PLOS Biology. This revised version of your manuscript has been evaluated by the PLOS Biology editors, the Academic Editor and the original reviewers, who are largely satisfied by the changes made.

Based on the reviews, we are likely to accept this manuscript for publication. However, before we can accept your manuscript, we encourage you to consider Reviewer 1's remaining remarks, and we suggest you consider incorporating the suggested references and points in your manuscript, as we think this would improve the discussion. 

**IMPORTANT: Additionally, as you consider reviewer 1's remaining points, please address the following data and other policy-related requests: 

1) TITLE: After some discussion, we think that the title should be edited to make it more broadly accessible. We think the ": a tale of double integration" is somewhat confusing to a reader who hasn't read your study, and so suggest that this either be removed, or possibly more clearly incorporated into the title to make it clear exactly what you did and/or found. For example, we might suggest the title be changed to something like (if you agree and if supported): 

--"Motor accumulation is a final arbiter in sensorimotor decision-making" 

or

--"A double integration, sensory evidence and motor accumulation, framework for understanding sensorimotor decision-making"

We will ultimately leave it up to you to refine the title further. 

2) ABSTRACT: Similar to the title, we think that the abstract should be edited to make the study more accessible to our broad readership. We have some specific suggestions to increase accessibility, but more broadly we suggest that it may be helpful for you to solicit input from a non-neuroscience colleague. 

--We think that it would be helpful to explain what a leaky motor accumulator is.

--We note you write about 'the behavioral data' and EEG signatures but dont explain this data - From the abstract, this could be a modelling study only- we suggest this be made clearer. 

-- We think that you should define what urgency manipulations are.

--Please note that per journal policy, the model system/species studied should be clearly stated in the abstract of your manuscript.

3) ETHICS STATEMENT: Thank you for providing an ethics statement in your methods section. Please update this to note whether the investigation was conducted according to the principles expressed in the Declaration of Helsinki.

4) BLURB: In the relevant section of our online system, please provide a blurb which (if accepted) will be included in our weekly and monthly Electronic Table of Contents, sent out to readers of PLOS Biology, and may be used to promote your article in social media. The blurb should be about 30-40 words long and is subject to editorial changes. It should, without exaggeration, entice people to read your manuscript. It should not be redundant with the title and should not contain acronyms or abbreviations. 

5) DATA AVAILABILITY: Thank you for providing the behavioral and EEG data related to your study on ownCloud. Is it possible to generate a DOI for this data? If not, we ask that you put this data on a different repository that allows for the generation of a permanent record. 

We expect to receive your revised manuscript within two weeks. 

*Published Peer Review History*

*Press*

Sincerely,

Luke

Lucas Smith, Ph.D.

Senior Editor,

PLOS Biology

lsmith@plos.org

Reviewer remarks:

Reviewer #1, Jeffrey D Schall (note: reviewer 1 signed this review): The authors have responded to the comments, critiques, and suggestions very well. I have no further comments to make about the revised manuscript, which is improved empirically and conceptually. However, in the spirit of scientific inquiry, let me offer some reactions to the comments of the other referees as they relate to this manuscript and associated work. Let me preface these remarks by stating that I am *not* seeking more citations, just better understanding.

R2C4a, the referee wrote: "It is generally believed that the decision threshold crossing precedes the onset of the motor response and that part of the "nondecision" time that typically is fitted when fitting standard integration-to threshold models is related to the preparation/execution of the motor response. Model signals that are time locked to the decision threshold crossing should therefore not directly be compared with neural signals that are time-locked to the onset of the motor response." - This comment is based on some architectural suppositions that recent work has addressed. Principally, the neurons doing the accumulation of evidence in the Gated Accumulator Model of Purcell et al. are the presaccadic movement neurons that can also be described as "motor preparation". In fact, in other work we showed that the neurons in FEF doing evidence accumulation are also doing response preparation and inhibtion: Middlebrooks PG, Zandbelt BB, Logan GD, Palmeri TJ, Schall JD. (2020) Countermanding perceptual decision-making. iScience 2020 23(1):100777. Because this manuscript is advancing the literature in its identification of distinguishable stages of processing, it seems useful to think carefully about the propositions linking particular neurons with particular processes. For what it's worth, I've meditated on this problem in this recent review: Schall JD (2019) Accumulators, neurons, and response time. Trends in Neurosciences. 42(12):848-860.

R2C5, in reply the authors wrote, "On the other hand, brain regions may not contain fully homogenous sets of neurons." - The heterogeneity of neurons in every cortical area has been established beyond dispute. We see it in FEF, e.g., Lowe Kaleb A. and Schall JD (2018) Functional categories of visuomotor neurons in macaque frontal eye field eNeuro 5(5): ENEURO.0131-18.2018. doi: 10.1523/ENEURO.0131-18.2018. It is also the case in LIP (e.g., Meister ML, Hennig JA, Huk AC. Signal multiplexing and single-neuron computations in lateral intraparietal area during decision-making. J Neurosci. 2013 Feb 6;33(6):2254-67), even though many authors do not incorporate such differences into their theoretical outlooks. 

Reviewer #2, Jochen Ditterich (note reviewer 2 signed this review): Thank you for your detailed response. All of my comments have been successfully addressed.

---

## [Editor Report · Decision Letter 3]

15 Jun 2023

Dear Dr Philiastides,

Thank you for the submission of your revised Research Article "Secondary motor integration as a final arbiter in sensorimotor decision-making" for publication in PLOS Biology and thank you for addressing our editorial requests in this revision. On behalf of my colleagues and the Academic Editor, Alexander Gail, I am pleased to say that we can in principle accept your manuscript for publication, provided you address any remaining formatting and reporting issues. These will be detailed in an email you should receive within 2-3 business days from our colleagues in the journal operations team; no action is required from you until then. Please note that we will not be able to formally accept your manuscript and schedule it for publication until you have completed any requested changes.

PRESS

Sincerely, 

Lucas Smith, Ph.D.

Senior Editor

PLOS Biology

lsmith@plos.org